# The oxytocin receptor gene polymorphism rs2268491 and serum oxytocin alterations are indicative of autism spectrum disorder: A case-control paediatric study in Iraq with personalized medicine implications

Zainab Al-Ali[1], Akeel Abed Yasseen[2], Arafat Al-Dujailli[3], Ahmed Jafar Al-Karaqully[4], Katherine Ann McAllister[5‡]*, Alaa Salah Jumaah[2‡]

1 Department of Pathology and Forensic Medicine, Faculty of Medicine, University of Kerbala, Kerbala, Kerbala Governorate, Iraq, 2 Department of Pathology and Forensic Medicine, Faculty of Medicine, University of Kufa, Kufa, Iraq, 3 Department of Internal Medicine, Faculty of Medicine, University of Kufa, Kufa, Najaf Governorate, Iraq, 4 Head of Psychiatric Department, Alhussain Teaching Hospital, Kerbala City, Kerbala Governorate, Iraq, 5 School of Biomedical Sciences, Ulster University, Northern Ireland, United Kingdom

‡ These authors are joint senior authors on this work.
* k.mcallister@ulster.ac.uk

## Abstract

### Background

Impairment of social functioning skills is a key hallmark of autism. The neuropeptide oxytocin (OXT) is a blood-based biomarker of social functioning, and a candidate for individualized treatment of ASD. The effects of OXT on the social brain are mediated by the OXT receptor (OXTR). This study assessed the clinical utility of blood OXT serum levels and the OXT receptor (OXTR) genotype as biomarkers of autism and its severity in a pediatric population in Iraq.

### Methods

Blood samples were collected from patients with a clinical diagnosis of ASD (n = 60) and corresponding age and gender matched healthy controls (n = 60). All clinical samples were processed at the Department of Pathology and Forensic Medicine, Faculty of Medicine, University of Kufa in Iraq. Blood serum was assayed for OXT by sandwich ELISA. Receiver operator analysis (ROC) determined area under the curve (AUC), cutoff values, and sensitivity and specificity of OXT values for accuracy of diagnosis of ASD. Isolated genomic DNA was genotyped for the OXTR gene rs2268491(C/T) SNP using allele-specific PCR. The significance of genotype (CC, CT, and TT) and allele (C and T) distributions in different patient groups was assessed using odd ratios (OR) with 95% confidence intervals (CI) and the Chi-square test. All statistical analysis was performed used SPSS software.

**Data Availability Statement:** All relevant data are within the paper and its Supporting Information files.

**Funding:** The author(s) received no specific funding for this work.

**Competing interests:** The authors have declared that no competing interests exist.

## Results

Study characteristics in the ASD population revealed a high level of consanguinity (36.66%), and ASD recurrence rate (11.66%) and family history (28.33%). OXT levels in patients with ASD (157.58±28.81 pg/ml) were significantly higher (p = 0.003) compared to controls (75.03±6.38 pg/ml). Within stratified ASD severity groups—OXT levels were significantly different (P = 0.032). ROC analysis determined similar AUC values for overall ASD (0.807), and stratified mild (0.793), moderate (0.889), and severe categories (0.795). The best cutoff for diagnosis of ASD was 83.8 pg/ml OXT with a sensitivity and specificity of 80% and 72.1% respectively. OXTR gene rs2268491(C/T) genotyping found that ASD patients have significantly lower (p = 0.021) genotype CC frequency and a significantly higher (p = 0.04) occurrence of the heterozygous CT genotype relative to controls. ASD subjects produced highest OXT levels with the TT genotype. T allele distribution was higher in ASD males. ASD males had significantly lower distribution of the CC genotype (48.89%) compared to females (80%) (Chi-square test: $\chi2$ = 4.43, df = 1, p = 0.035). Whereas distribution of the CT genotype was significantly higher in autistic males (44.45%) compared to females (13.33%) (Chi-square test: $\chi2$ = 4.68, df = 1, p = 0.03).

## Conclusion

Peripheral OXT levels and OXTR genetic alterations are potential biomarkers of social functioning in the ASD patient setting. The stratification of patients with ASD into severity categories shows significant differences both in OXT levels and OXTR (rs2268491, C/T) genotype and allele distributions, that can be sex dependent. OXT based therapies will require personalized medicine tactics to correctly identify patients with ASD who require neuropeptide boosting in social settings.

## Introduction

Autism spectrum disorder (ASD) is a neurodevelopmental disorder characterized by early age social communication deficits, language delay and repetitive sensory motor behaviors. The clinical presentation of autism is usually lifelong, while severity ranges from very mild in some people to a severe occurrence of developmental disability in others [1]. Autism is not a single clinical disorder because of this variable phenotypic spectrum. Substantial care is required to support individuals with ASD who cannot function independently. This causes an economic burden for society. The worldwide elevation in autism stresses the need for collective global efforts to investigate this problem to support autistic children and their families. The prevalence of autism has increased in developed countries [2], with recent estimates of 1.70% and 1.85% in US children aged 4 and 8 years respectively [3]. The pathogenesis of ASD is still poorly understood, and is believed to entail genetic and environmental risk factors [4].

The incidence of ASD in Middle East countries has increased. Both the volatile environment in parts of the Middle East and genetic factors could exacerbate the development of autism. However there is a paucity of genetic and molecular biology studies among these populations to provide adequate diagnostic methods, patient support and guidance [5]. Twin and family research studies provide compelling evidence for the role of genetics in the occurrence of autism. Family studies reveal an eight to ten percent recurrence risk of autism in the siblings

of affected probands [6]. Blood-related marriages may increase the risk of producing offspring with ASD [7]. While consanguinity occurs in 10.4% of the global population [8], Arab countries have the highest rates approaching 60% [9]. Therefore evaluation and diagnosis of ASD in these communities should entail screening for consanguinity.

Oxytocin is a potential diagnostic metric for the social core symptoms of ASD. The plasma levels of OXT are positively correlated with autistic diagnostic interview (ADI) reciprocal interaction and communication scores [10]. However studies report contradictory blood levels of OXT in autistic individuals that compromise its clinical utility. Lower OXT levels have been reported in European (Slovakia), US, and Middle East (Iraq) populations of children with ASD in comparison with normal healthy controls [10–12]. One study in Germany [13] reported no significant difference in OXT levels between ASD and healthy control patients (p = 0.132). Whereas other studies report elevated plasma OXT level among US and European children and adults with ASD [14, 15] and Chinese Han children with ASD [16] relative to healthy controls. The association between OXT levels and social function in ASD is not straightforward and requires further investigation to determine whether high or low serum OXT levels are a biomarker of ASD.

Single nucleotide polymorphisms (SNPs) in the OXT receptor (OXTR) gene are associated with autism, and include rs2268491(C/T), rs2254298 (G/A) and rs53576 (G/A). Recently brain activity was found markedly reduced in adolescent autistic females with the rs2268491 genotype who responded to an emotion recognition task in comparison to normal controls [17]. Furthermore the OXTR SNPs rs2254298 (G/A) and rs53576 (G/A) are found associated with autism in the Chinese Han population [18] and rs2254298 in Caucasian patients [19]. Conversely meta-analysis suggests no association of both rs2254298 (G/A) and rs53576 (G/A) SNPs with human social behaviour [20].

Building upon all the above arguments, we investigated the OXTR serum levels and OXTR SNP RS2268491 in a pediatric population with ASD in Iraq. Our objectives were to (1) determine if serum OXT concentration could be used as an ASD biomarker (2) determine genotype and allele distributions of rs2268491 in patients with ASD (3) explore associations between OXT protein and the OXTR polymorphism.

## Materials and methods

### Study population and sample size calculation

The study was conducted at the Department of Pathology and Forensic Medicine, Faculty of Medicine, University of Kufa. During a one-year period (December 2019 to December 2020) paediatric patients (median age of 7, range 3–15 years) with a clinical diagnosis of ASD by the study consultant psychiatrist were recruited.

A sample size of N = 36 was calculated as the minimal requirement to determine serum oxytocin alterations in the study. The study variable of interest (peripheral oxytocin alterations) was identified as dichotomous (proportion). The sample size was calculated using the following equation: $N = 4 * Z\alpha2 * p (1 - p)/w^2$. Where Z$\alpha$ is the confidence level, W is the width of the confidence interval (equal to twice the margin of error) and P is the study estimate of the proportion to be measured [21]. The proportion of ASD patients who present with oxytocin alterations was estimated at 90% compared to controls (S1 Appendix), with a selected confidence interval of ± 10 [21]. $N = 4 * 1.96 * 0.9 (1 - 0.9)/0.2^2 = 17.64$ (18 cases of ASD)

Minimum sample size for 1:1 case-control study: N = 36

A final total of 60 cases of ASD were recruited for the study along with age and gender matched healthy controls (n = 60) for comparison. All the cases were recruited from the teaching hospitals scattered throughout the Middle and South Euphrates regions in Iraq.

## Inclusion and exclusion criteria, CARS evaluation of pediatric ASD

Patients who fulfilled the criteria of being a typical ASD subject according to the diagnostic criteria of the Diagnostic and Statistical Manual of Mental Disorders, (American Psychiatric Association) were included [22]. The diagnosis of ASD was confirmed by a consultant psychiatrist who evaluated the child in person, in collaboration with the research team. The patients with ASD were stratified according to severity into the following sub-groups [22]: mild ASD (n = 39), moderate ASD (n = 13) and severe ASD (n = 8) according to the Childhood Autism Rating Scale (CARS) [23]. The CARS is a diagnostic assessment method that rated individuals on a scale ranging from normal to severe and yields a composite score ranging from non-autistic to mildly autistic, moderately autistic, or severely autistic. The scale was used to observe and subjectively rate the following fifteen items in both ASD and case control subjects. Each of the following 15-items were scored in patients according to seven levels of severity by interview:

1. relationship to people

2. imitation

3. emotional response

4. body

5. object use

6. adaptation to change

7. visual response

8. listening response

9. taste-smell-touch response and use

10. fear and nervousness

11. verbal communication

12. non-verbal communication

13. activity level

14. level and consistency of intellectual response

15. general impressions

The controls were normal healthy children, unrelated to the autistic subjects and without any of the following exclusion criteria:

1. Any medical condition likely to be etiological for ASD (e.g. Rett syndrome, focal epilepsy).

2. Any neurologic disorder involving pathology above the brain stem, other than uncomplicated non-focal epilepsy.

3. Contemporaneous evidence, or unequivocal retrospective evidence, of probable neonatal brain damage.

4. Any genetic syndrome involving the CNS, even if the link with autism is uncertain.

5. Clinically significant visual or auditory impairment, even after correction.

6. Any circumstances that might possibly account for the picture of autism (e.g. severe nutritional or psychological deprivation).

7. Active treatment with pharmacological or other agents.

## Blood sample collection and preparation

A venous blood sample was collected from patients with ASD and healthy controls in parallel at the University of Kufa teaching hospital, Department of Clinical Laboratories. Both groups were matched in both gender and age. A 3.0 mL whole blood volume sample was transferred to a sterilize serum collection tube and allowed to clot for 20 minutes at room temperature. This sample was centrifuged at 2000–3000 RPM for 20 minutes. The separated serum was divided into small aliquots and stored at -20°C for later OXT measurements. A further 2.5 mL of whole blood was transferred into a EDTA tube for DNA extraction and PCR genotyping. All laboratory work was performed at the Department of Pathology and Forensic Medicine, Faculty of Medicine, University of Kufa.

## Isolation of genomic DNA for PCR genotyping

Instruments and equipment, along with chemicals and reagents for DNA extraction and PCR are available in S1 and S2 Tables.

Genomic DNA was isolated from 2.5 mL whole blood samples using the Column-pure blood Genomic DNA Mini Kit, Applied Biological Materials (Anatolia Turkey) according to the manufacturer instructions under sterile conditions (S3 Table). Genomic DNA concentrations were measured using the Nano-drop spectrophotometer. The DNA concentrations were determined by measuring the absorbance at 260 nm wavelength (A260) and 280 nm wavelength (A280). Purity was determined by calculating the ratio of absorbance at 260 nm and the absorbance at 280 nm (A260/A280). Absorbance scans showed a symmetric peak at 260 nm confirming high purity. The purity of DNA -was considered acceptable if it was in the range of 1.8–2.0.

DNA quality was assessed by gel electrophoresis (S3 Table). The electrophoresis process was conducted at 5–8 voltage/cm for 45 min. After termination of electrophoresis, the agarose gel was visualized using a UV-transilluminator (Cleaver Scientific Co., UK) [24]. Agarose gel electrophoresis was used in this study to fulfill two purposes: (1) evaluate the quality of genomic DNA samples enrolled in the study prior to allele specific PCR and (2) check the presence of PCR products at the expected sizes after termination of the allele specific PCR.

## Allele specific PCR primer design and synthesis

The study cohort DNA samples were genotyped for OXTR SNP accession numbers rs2268491 using the allele specific PCR technique.

All allele specific PCR primers used in this study were designed manually. The SNP was retrieved from the database dbSNP (contains human SNP variations, microsatellites, small scale-insertions, and deletions, https://www.ncbi.nlm.nih.gov/snp/). Mainly nucleotide sequences of 1000 bp or 500bp containing the SNP were retrieved from the database. Then, the allele specific primers were designed manually and were computationally checked regarding 3' complementarity, 3'self -complementarity, GC content, and melting temperature using the primer-blast online program, localized at the server (https://www.ncbi.nlm.nih.gov/tools/primer-blast/) from NCBI (National Center for Biotechnology and Information). All designed primers in this study were synthesized in Integrated DNA Biotechnology (IDT Co., Canada).

The sequences of allele specific primers for the oxytocin receptor SNP gene rs2268491, and full PCR cycling condition are outlined in S4 Table.

Effective SNP discrimination by allele-specific PCR was performed using standard PCR conditions. Each DNA sample (1 μg/PCR reaction) was processed through two alleles specific PCR reactions. Each allele specific PCR reaction was directed with a primer set: forward primer carrying the allele SNP base at 3' prime end and a reverse primer. The control reaction was established like the allele specific PCR reaction, except that the forward allele primer was replaced by control forward primer (primer set: control forward and common reverse primer). Two allele-specific PCR products were generated of different lengths and separated by agarose gel electrophoresis for direct visualization of the genotyping result.

## Enzyme-Linked Immunosorbent Assay (ELISA) oxytocin measurements

The ELISA quantitative immunoassay sandwich kit measured human OXT in serum (manufacturer, Bioassay Technology laboratory). The assay range was 2 pg/ml to 600 pg/ml. Each serum sample was added in duplicate to a pre-coated human OXT antibody plate. Biotinylated human OT Antibody was added to bind to OXT in the sample followed by streptavidin-HRP. After incubation, unbound Streptavidin-HRP is washed away. The substrate solution was added to enable color development in proportion to the amount of human OXT. The reaction was terminated using acidic stop solution and absorbance measured at 450 nm with a plate reader (Bioassay Technology laboratory) and serum OXT concentration assessed via the standard curve method.

## Statistical analysis

The Statistical Package of Social Sciences version 27 (SPSS Inc.; Chicago, IL, USA) computer program was used for results analysis. For each variable the values were presented as mean ± SD. The student t test determined the statistical difference between two groups. One-Way ANOVA was performed to evaluate the differences among multiple groups. For statistical comparison between different groups, the statistical significance was defined as $p \leq 0.05$, while a p-value of $>0.05$ was not significant. A p-value of $<0.001$ was considered highly significant. In the studied groups, the representativeness of alleles and genotypes was estimated by the Hardy-Weinberg equilibrium (HWE) by comparing the observed and expected frequencies of genetic variants. The Chi-square test was applied to assess genotype and allele frequencies between patients and controls. The genotype and allele distributions were determined in each group, and odds ratios (OR) with 95% confidence intervals (95% CI) were calculated. A p-value of $<0.05$ was considered statistically significant at a confidence interval (CI) of 95%. Receiver operating characteristic (ROC) analysis assessed the accuracy of OXT as a biomarker for autism. The area under the curve (AUC) indicates an excellent diagnostic and predictive marker when close to one, a curve that lies close to the diagonal (AUC = 0.5) has no diagnostic significance. Normality testing of OXT levels in the patient serum samples was assessed using the Kolmogorov-Smirnov test.

## Ethical approval

The present study was approved by the Institutional Review Board of the University of Kufa, Faculty of Medicine, in accordance with the 1964 Helsinki declaration and its later amendments. "The authors are accountable for all aspects of the work in ensuring that questions related to the accuracy or integrity of any part of the work are appropriately investigated and resolved." All patients were informed about the purpose of the work and written parental consent and questionnaire information was obtained from the father of each patient.

## Results

### Study patient characteristics

The patients with ASD had a mean age of 7.08± 2.54 compared to the control group 6.95±1.94 (Table 1). There were no significant differences in age between both groups (p = 0.708). The rate of consanguinity among the Arabic study populations was high at 36.66% and 40% in the ASD and healthy control groups. Within the stratified population of ASD subjects, the consanguinity rates occurred at 35.89%, 30.76% and 50% in mild, moderate, and severe cases. Recurrence of ASD within the same family occurred in 7 cases (11.66%) out of the total population (n = 60). Furthermore 28.33% of the ASD population had a family history of the condition, that increased in the most severe cases to 37.50%. Eight patients (13.3%) out of the population with ASD suffered from fit episodes, that occurred at higher frequency in the severe cases (37.5%).

### Serum OXT levels in ASD and control subjects and ROC curve analysis

Table 2 shows the ELISA measurements of serum OXT among the ASD patients (n = 60) and the healthy control group (n = 60) in the present study. The normal distribution of the results was checked using Kolmogorov-Smirnov testing (p>0.05). Serum OXT levels in the patients with ASD (157.58±28.81 pg/ml) were significantly higher (p = 0.003) in comparison to the healthy control group (75.03±6.38 pg/ml). The serum OXT levels in the patients with ASD were stratified into severity categories of mild (n = 39), moderate (n = 13), and severe (n = 8). The ANOVA test found statistically significant difference (p = 0.032) in OXT levels among the ASD severity groups, as shown in Table 2. The greatest significance difference in OXT occurred between mild and severe ASD patients (p = 0.001). The correlation of OXT with age in ASD patients was not significant (P = 0.396, R = -0.140, $R^2$ = 0.0196; S1 Fig). There was no also significant difference (p>0.05) in OXT levels in children from related parents compared to unrelated (S5 Table). Fig 1A shows the elevated OXT levels in the overall ASD population and stratified ASD severe subtype in comparison to lower levels in healthy controls.

Table 3 show the ROC analysis of OXT serum data presented as AUC, best cut off values, specificity, and sensitivity in patient groups. Fig 1B shows the ROC curve for all ASD cases,

**Table 1. Patient ethnicity, consanguinity, recurrence, and family history of ASD.**

| Characteristics | Control patients | ASD patients | Mild ASD | Moderate ASD | Severe ASD |
|---|---|---|---|---|---|
| N | 60 | 60 | 39 | 13 | 8 |
| Ethnicity | Arab | Arab | Arab | Arab | Arab |
| Mean Age (+/-SD) | [a]6.95 ± 1.94 | 7.08± 2.54 | 6.89±2.19 | 7.23±3.81 | 7.78±3.67 |
| Age Range, Median | 3–15, 7 | 3–15, 7 | 3–15, 7 | 3–15, 10 | 4–15, 8 |
| Consanguanity | 24 (40.0) | 22 (36.66) | 14 (35.89) | 4 (30.76) | 4 (50) |
| OR | - | 1 | 0.97 | 0.77 | 1.73 |
| P value | - | 1 | 0.94 | 0.69 | 0.47 |
| 95% CI | - | 0.48–2.10 | 0.42–2.24 | 0.21–2.79 | 0.39–7.60 |
| Recurrent (Same family) | 0 (0) | 7 (11.66) | 6 (15.38) | 1 (7.69) | 0 (0) |
| ASD Family History | 0 (0) | 17 (28.33) | 12 (30.76) | 3 (23.07) | 3 (37.50) |
| [b]Fit | 0 (0) | 8 (13.33) | 2 (5.12) | 3 (23.07) | 3 (37.5) |

[a]No significant differences in age between control and ASD groups (p = 0.708).

[b]Fit is a convulsion or seizure and a clinical sign that occurs when there is a sudden burst of electrical activity in the brain temporarily interfering with the normal messaging processes.

**Table 2. ELISA serum OXT levels among study participants.**

| Group | N | [a]Serum OXT (pg/mL) | P-value OXT |
|---|---|---|---|
| Control Patients | 60 | 75.03±6.38 | 0.003* |
| ASD Patients | 60 | 157.58±28.81 | |
| Mild ASD | 39 | 165.02±28.03 | [c]0.032* |
| Moderate ASD | 13 | 152.78±27.86 | |
| Severe ASD | 8 | 129.09±11.04 | |

[a]Normality tested using Kolmogorov-Smirnov test: controls (0.055), ASD (0.072), mild ASD (0.21), moderate ASD (0.2), severe ASD (0.2).

[b]No significant differences in age between control and ASD groups (p = 0.708).

[c]ANOVA test compared OXT levels among the ASD severity groups of mild, moderate, and severe (p = 0.032*). Significance was also compared between specific ASD severity groups: mild versus moderate (p = 0.178), mild versus severe (p = 0.001*), and moderate versus severe (0.034*).

*Denotes a significant difference (*P*<0.05).

while Fig 1C–1E shows the mild, moderate, and severe groups independently. The OXT exhibited AUC values higher than 0.8 in the overall ASD patient population (AUC = 0.807), and similar values in mild (AUC = 0.793) moderate (AUC = 0.889), and severe autistic patients (AUC = 0.795). The best cutoff in pediatric serum samples of 83.8 pg/mL OXT offers a diagnostic accuracy of ASD at sensitivity and specificity levels of 80% and 72.1% respectively. Within the stratified population of ASD, the cutoff was highest in mild patients (87.2 pg/mL) and lowest in severe cases (71.4 pg/mL). The highest test accuracy occurred in patient with moderate ASD according to sensitivity and specificity indicators.

## OXTR gene rs2268491 polymorphism in study population

Table 4 reports the allele and genotype distributions of the OXTR SNP rs2268491 (C, T) in the study population. ASD patients had a significantly lower (p = 0.021) frequency of the wild-

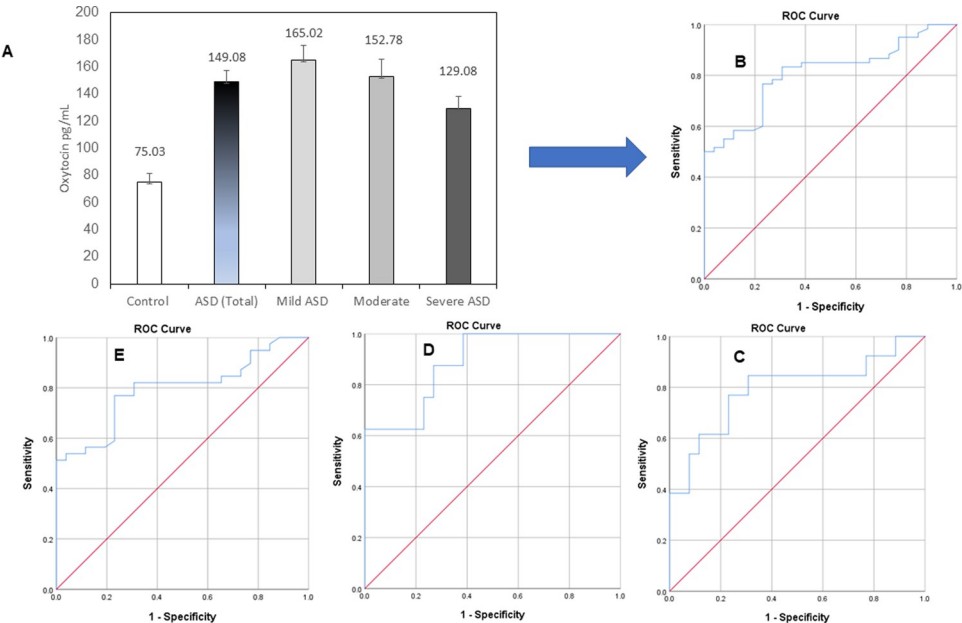

**Fig 1.** **(A)** Oxytocin levels among study participants, **(B)** ROC curve analysis in all ASD cases, **(C)** ROC curve analysis in mild ASD, **(D)** ROC curve analysis in moderate ASD, **(E)** ROC curve analysis in severe ASD.

**Table 3. ROC analysis.**

| Groups | Area under curve | Best cutoff (pg/mL) | Sensitivity | Specificity |
|---|---|---|---|---|
| ASD patients | 0.807 | 83.8 | 80 | 72.1 |
| Mild | 0.793 | 87.2 | 76 | 68.2 |
| Moderate | 0.889 | 78.3 | 86 | 80.6 |
| Severe | 0.795 | 71.4 | 75 | 70.4 |

**Table 4. Genotype distribution (CC, CT, TT) and alleles (C, T) of rs2268491 in patients with ASD (N = 60) and healthy controls (N = 60).**

| Genotype / Allele | Patients | | Control | | P value | OR | 95% CI |
|---|---|---|---|---|---|---|---|
| | No. | % | No. | % | | | |
| Genotype | | | | | | | |
| CC | 34 | 56.67 | 46 | 76.66 | 0.021* | 0.398 | 0.1813 to 0.8739 |
| CT | 22 | 36.67 | 12 | 20 | 0.04* | 2.315 | 1.0178 to 5.2692 |
| TT | 4 | 6.66 | 2 | 3.33 | 0.411 | 2.071 | 0.3648 to 11.762 |
| Alleles | | | | | | | |
| C | 90 | 75 | 104 | 86.67 | 0.866 | 1 | 0.5418 to 1.8457 |
| T | 30 | 25 | 16 | 13.33 | 0.866 | 1 | 0.5418 to 1.8457 |

*Denotes a significant difference ($P<0.05$).

type genotype CC (56.67% versus 76.66%) and a significantly higher (p = 0.04) occurrence of the heterozygous CT genotype (36.67% versus 20%) relative to healthy controls. The odd ratio (OR) for both the CT and the TT genotype favored the ASD population at 2.315 (95%CI; 1.0178–5.2692) and 2.071 (95%CI; 0.3648–11.762) respectively. Whereas likelihood of the wild-type CC homozygous genotype is reduced in the ASD population with an OR of 0.398 (95% CI; 0.1813–0.8739) and favors the healthy population. The C allele distribution was higher in healthy controls while the T allele distribution was higher in the ASD population.

The distribution of the OXTR SNP rs2254298 genotypes according to the Hardy-Weinberg equilibrium found consistency in both groups ($\chi 2 = 0.03$, df = 1, p >0.05 in ASD subjects; $\chi 2 = 0.01$, df = 1, p >0.05 in healthy controls).

## OXTR gene rs2268491 polymorphism in ASD males and females

Table 5 reports the allele and genotype distributions of the OXTR SNP rs2268491 (C, T) in males and females of the ASD study population. The distribution of the CC genotype was

**Table 5. OXTR polymorphism rs2268491 in ASD males (n = 45) and females (n = 15).**

| Genotype / allele | Male | | Female | | DF | $X^2$ | P-value |
|---|---|---|---|---|---|---|---|
| | No. | % | No. | % | | | |
| Genotype | | | | | | | |
| CC | 22 | 48.89 | 12 | 80 | 1 | 4.43 | 0.035* |
| CT | 20 | 44.45 | 2 | 13.33 | 1 | 4.68 | 0.03* |
| TT | 3 | 6.66 | 1 | 6.66 | 1 | 0 | 1 |
| Alleles | | | | | | | |
| C | 64 | 71.12 | 26 | 86.67 | 1 | 2.904 | 0.088 |
| T | 26 | 28.88 | 4 | 13.33 | 1 | 2.904 | 0.088 |

*Significant association ($P<0.05$).

**Table 6. OXTR polymorphism rs2268491 in ASD severity categories.**

| Genotype / Allele | Mild | | Moderate | | Severe | | DF | $X^2$ | P -value |
|---|---|---|---|---|---|---|---|---|---|
| | No. | % | No. | % | No. | % | | | |
| Genotype | | | | | | | | | |
| CC | 20 | 51.29 | 8 | 61.5 | 6 | 75 | 2 | 1.68 | 0.431 |
| CT | 16 | 41.0 | 4 | 30.7 | 2 | 25 | 2 | 0.98 | 0.612 |
| TT | 3 | 7.69 | 1 | 7.69 | 0 | 0 | 2 | 0.65 | 0.719 |
| Alleles | | | | | | | | | |
| C | 56 | 71.7 | 20 | 76.9 | 14 | 87.5 | 2 | 1.81 | 0.404 |
| T | 22 | 28.2 | 6 | 23.07 | 2 | 12.5 | 2 | 1.81 | 0.404 |

significantly lower in males (48.89%) compared to females (80%) according to the Chi-square test ($\chi$2 = 4.43, df = 1, p = 0.035). Whereas distribution of the CT genotype was significantly higher in autistic males (44.45%) compared to females (13.33%) using the Chi-square test ($\chi$2 = 4.68, df = 1, p = 0.03). No significant difference was evidenced in the distribution of the genotype TT among the male and female autistic patients. Overall the C allele distribution was higher in females while the T allele distribution was higher in males.

## OXTR gene rs2268491 polymorphism in ASD severity categories

Table 6 reports the allele and genotype distributions of the OXTR SNP rs2268491 (C, T) in the ASD mild (n = 39), moderate (n = 13) and severe (n = 8) cases. There are no significant findings in the genotype or allele frequencies in any of the severity categories.

## OXT levels according to genotype distributions in stratified autism patients and healthy controls

The OXT levels in stratified autistic and control groups are presented according to genotype distribution of rs2268491 in Table 7 and Fig 2. A marked trend for higher OXT was observed in the three genotypes of CC, CT, and TT in ASD patients relative to control. The healthy controls produced highest OXT protein (85.45±8.01 pg/mL) in the normal homozygous CC genotype, however there are no significant differences between the three control genotypes (CC, CT, TT). ASD subjects produced the highest OXT in the TT genotype (169.52±8.42 pg/mL). The ASD subjects exhibited a significant association between serum oxytocin levels and the three OXTR SNP rs2254298 genotypes (p = 0.045). In stratified ASD patients, serum oxytocin levels were highest in the mild category with the polymorphism genotype TT (175.02±13.05). The moderate ASD cases displayed significant differences among the three genotypes in serum

**Table 7. OXTR polymorphism rs2268491 in ASD severity categories.**

| Group | Genotype OXT levels (mean pg/ml ±SD) | | | P value |
|---|---|---|---|---|
| | CC | CT | TT | |
| Control | 85.45±8.01 | 70.73±9.18 | 68.92±7.24 | 0.056 |
| ASD | 145.21±9.21 | 132.51±5.08 | 169.52±8.42 | 0.045* |
| Mild | 161.74±12.04 | 158.12±14.2 | 175.02±13.05 | 0.118 |
| Moderate | 139.34±8.21 | 114.41±4.8 | 164.01±9.05 | 0.038* |
| Severe | 134.54±4.28 | 125.02±8.72 | —————— | 0.069 |

*Significant association (P<0.05).

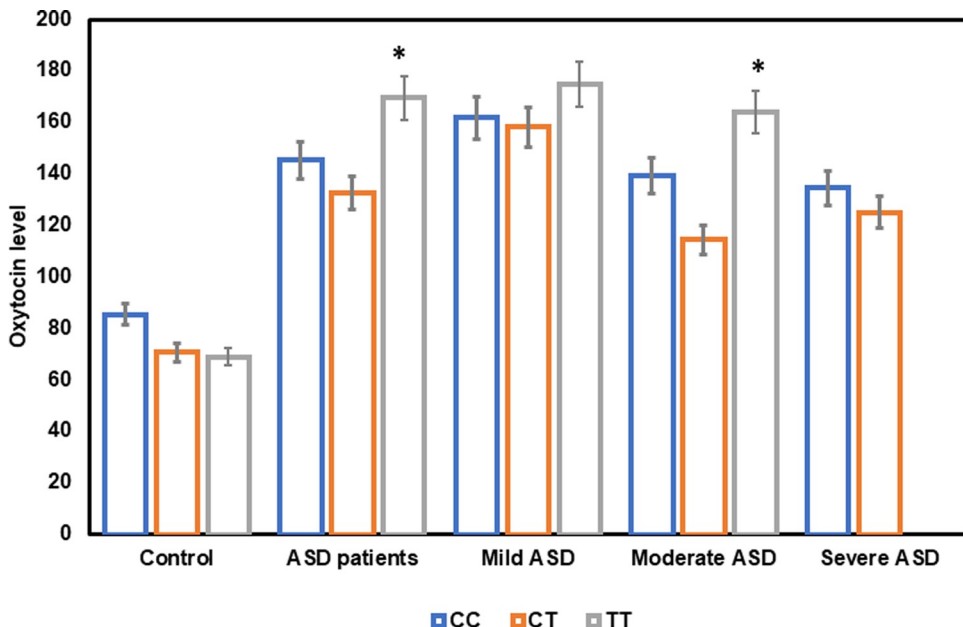

**Fig 2. Bar chart of the oxytocin levels according to the three oxytocin gene polymorphism genotypes in autism patients and controls.**

OXT levels (p = 0.038). The ASD severe stratified population had lower numbers (n = 8) reducing power of interpreting observations.

## Discussion

This study aimed to develop a framework to improve diagnostic testing of children and adolescents with ASD in Iraq to help improve their outlook as adults. There is a currently a paucity of genetic and molecular biology studies of these specific populations to provide adequate diagnosis, to support patients and parents and issue guidance for the future [4]. Our study characteristics revealed a high level of consanguinity (36.66%), recurrence rate (11.66%) and family history (28.33%) among the ASD population. Given that the consanguinity rate worldwide is 10.4% [8], the rates observed in both the ASD and control study populations are at least three fold higher. Therefore it is likely that specific gene pools are the root cause of some of the ASD cases in the study population of Arabs. The ASD candidate genes need to be identified so that a personalized medicine treatment plan can be developed for future pediatric ASD cases to improve their prognosis and social functioning to achieve independency within society as adults.

Autism is a lifelong neurodevelopmental disorder. However studies show that early diagnosis and intervention leads to significantly improved outcomes. Ignoring early and accurate diagnosis of this disorder might lead to secondary disorders such as depression and anxiety [25]. Currently there is no gold standard treatment to improve social functioning skills in autism, apart from medications that help to relieve the comorbid anxiety and panic disorders associated with the condition. Since the neuropeptide OXT and its receptor OXTR regulate social functioning in animals and humans, it is the front-runner among candidate social treatments for ASD. Plasma OXT concentrations and OXTR polymorphisms predict social impairments in children with and without ASD [26]. Furthermore intranasal OXT administration improves social abilities in children with ASD [26]. The probability that innovative uses of

OXT will improve the lives of autism patients is twofold because of its dual therapeutic and diagnostic potential.

Our study found that serum OXT levels are significantly elevated (p = 0.003) in the pediatric patients with ASD in comparison to the matched healthy control group. The gold standard statistical tool of ROC analysis was used to demonstrate the potential of OXT as a blood-based biomarker of ASD. OXT had excellent AUC values higher than 0.8 in the overall ASD patient population (AUC = 0.807), and along with mild (AUC = 0.793) moderate (AUC = 0.889), and severe autistic patients (AUC = 0.795) and had good accuracy with specificity and sensitivity values. These results prove the potential role of OXT serum measurements in the diagnosis of ASD at optimal cut off values, assessment of severity, and, for prognostic purposes in this specific ethnic population.

The trend for elevated OXT in this study agrees with other studies [14–16]. However, the use of OXT as a biomarker of ASD is not straightforward, as these results are contradictory with those reporting lower OXT in ASD patients [10–12]. However, within the stratified ASD population in this study, the highest OXT levels occurred in the mild subgroup, while the lowest OXT levels occurred in the severe group. Our study found that ASD severity categories had statistically significant (p = 0.032) difference in OXT levels. This trend does agree with the previous study of ASD children in Iraq [12]. Here OXT level were highest in mild autistic patients with a significant decrease (p<0.05) in moderate autistic patients, and a highly significant decrease (p<0.01) in severe autistic patients compared with control. Interestingly, this variability in OXT levels in stratified ASD patients reported both Iraqi studies may relate to a key finding of the Stanford study [27]. The US investigators noted that pretreatment blood OXT concentrations also predicted response to intranasal OXT treatment [27]. Those individuals with the lowest pretreatment OXT concentrations showed the greatest social improvement [27]. Therefore, the ASD severe patients with lowest OXT in our study could be suitable candidates for individualized intranasal therapy with OXT to boost their social functioning. However, much wider study populations and statistical analysis is required to reach any conclusion before OXT concentrations are used as hallmarks for ASD diagnosis and treatments.

Emerging evidence also links alterations in OXT signaling pathways and its receptor -the OXTR, in the etiology of ASD. The effects of OXT on the social brain are mediated by the OXTR. A recent study used *in vivo* arterial spin labelling to identify changes in cerebral blood flow following intranasal administration of OXT that implicated many target areas within the brain [28]. The knockout of OXT receptors in mouse models showed autistic-like deficits in social interaction [29].

The OXTR SNP rs2268491 has been significantly associated with ASD by meta-analysis [30]. In particular, T allele carriers of the SNP are strongly linked to ASD related social behaviors [30]. Brain imaging also shows that activity is markedly reduced in adolescent autistic T carriers of the rs2268491 genotype when given a social decision task in comparison to normal control T carriers [17]. Our SNP genotyping study (rs2268491) discovered that ASD patients had a significantly lower (p = 0.021) frequency of the genotype CC and a significantly higher (p = 0.04) occurrence of the heterozygous CT genotype relative to healthy controls. This finding again suggests that the T allele is contributing to ASD.

Our study also found that ASD subjects with the TT genotype had the highest OXT levels. The elevated OXT levels noted among these patients are likely a reflection of genetic changes in the OXTR rs2254298 SNP. The ASD subjects exhibited a significant association between serum oxytocin levels and the three OXTR SNP rs2254298 genotypes (p = 0.045). In stratified ASD patients, serum OXT levels were highest in the mild category with the polymorphism genotype TT. This also suggests that patients with milder ASD symptoms and the TT genotype may have evolved an ability to upregulate OXT levels to compensate for the cerebral social

response skills deficiency caused by the OXTR polymorphism. The higher amounts of OXT will help to improve facial processing and human interpersonal contact, causing milder symptoms of ASD. A recent mouse ASD model study has demonstrated that targeting OXTR-expressing neurons in the lateral septum restores social skills [31]. Both our current study and previous research [27, 31] suggest that OXT is adaptive and can restore the homeostasis in the body.

Autism also has a sex bias with up to five times higher frequency amongst boys compared to girls [32]. There are also known sex differences in the phenotypic presentation of ASD in girls compared to boys. Our study found that males with ASD had a significantly lower distribution of the normal homozygous CC genotype (48.89%) compared to females (80%) (Chi-square test: $\chi2 = 4.43$, df = 1, p = 0.035). Whereas distribution of the CT genotype was significantly higher in autistic males (44.45%) compared to females (13.33%) (Chi-square test: $\chi2 = 4.68$, df = 1, p = 0.03). The T allele distribution was more than twice as high in male subjects. Given that the T allele carriers have compromised OXT mediated brain activity for social responses, this could provide a genetic explanation for why ASD affects some of the girls less frequently and possibly less severely in phenotypic presentation than boys. There may be an underlying OXT related mechanism impaired more so in the boys that gives rise to greater preponderance of ASD.

The implications of this study are that measurement of OXT levels and genetic alterations in its receptor could serve as a simple blood-based biomarker test to gauge social functioning in the ASD patient for diagnostic, monitoring and support purposes. Genetic and protein biomarkers in peripheral blood samples are easier and less expensive to analyze with a high throughput potential in the laboratory for ASD diagnostics, in comparison with genome-wide sequencing or brain imaging approaches.

## Study limitations

Ideally the conclusions presented in this study should be generalizable to the wider Iraqi Arab population. Therefore the study samples must be representative of the population and adequate in number [33]. Feasibility issues in the Middle and South Euphrates population region limited recruitment of large sample sizes in the current study. Although the estimated study sample size was sufficient for the serum OXT analysis, numbers are still inadequate to generalize to the entire population. For example, consanguinity was high among the ASD populations as expected, but also the control group. The sample size might explain why the consanguinity proportions are higher among control patients. Sample size also affects the association between SNP markers and disease. Testing a single SNP marker in a 1:1 case control study requires 248 cases to achieve 80% statistical power according to the allelic genetic model, under certain assumptions [34]. Insufficient study numbers are a limitation of the current investigation of SNP rs2254298.

## Conclusion

In conclusion the stratification of patients with ASD into severity categories shows statistically significant (p = 0.032) differences in OXT levels. The elevated OXT levels noted among patients with ASD are likely a reflection of genetic changes in the OXTR, especially when certain genotype or alleles are more predominant in younger patients with ASD. Certain OXTR genotypes such as the T carriers of rs2268491 may have upregulated levels of peripheral OXT to compensate for ineffective processing of OXT at its receptor for social functioning in ASD. Undoubtably, the successful application of OXT based therapies for ASD patients will

necessitate personalized medicine tactics to stratify patients who require neuropeptide boosting in social settings.

## Supporting information

**S1 Fig. Nonsignificant correlation of serum OCT levels and age of ASD patient cohort (P = 0.396, R = -0.140, $R^2$ = 0.0196).**
(TIF)

**S1 Table. Instruments and equipment used in the study.**
(DOCX)

**S2 Table. Chemicals and reagents for DNA extraction and polymerase chain reaction.**
(DOCX)

**S3 Table. Lab protocols.**
(DOCX)

**S4 Table. The sequences of allele specific primers of oxytocin receptor diallelic gene rs2268491 (C/T) and PCR information.**
(DOCX)

**S5 Table. OXT levels of control and autistic groups with related and unrelated parents.**
(DOCX)

**S1 Appendix. Study datasheet.**
(XLSX)

## Acknowledgments

The study participant children and their parents who supported this study are gratefully acknowledged by all the study authors.

## Author Contributions

**Conceptualization:** Akeel Abed Yasseen, Katherine Ann McAllister.

**Data curation:** Zainab Al-Ali.

**Formal analysis:** Zainab Al-Ali, Akeel Abed Yasseen, Katherine Ann McAllister.

**Funding acquisition:** Akeel Abed Yasseen.

**Investigation:** Zainab Al-Ali, Akeel Abed Yasseen, Arafat Al-Dujailli, Ahmed Jafar Al-Karaqully, Alaa Salah Jumaah.

**Methodology:** Zainab Al-Ali, Akeel Abed Yasseen, Arafat Al-Dujailli, Ahmed Jafar Al-Karaqully, Katherine Ann McAllister, Alaa Salah Jumaah.

**Project administration:** Akeel Abed Yasseen, Arafat Al-Dujailli, Ahmed Jafar Al-Karaqully, Alaa Salah Jumaah.

**Resources:** Akeel Abed Yasseen, Arafat Al-Dujailli, Ahmed Jafar Al-Karaqully.

**Software:** Zainab Al-Ali, Alaa Salah Jumaah.

**Supervision:** Akeel Abed Yasseen, Alaa Salah Jumaah.

**Validation:** Akeel Abed Yasseen, Alaa Salah Jumaah.

**Visualization:** Zainab Al-Ali.

**Writing – original draft:** Zainab Al-Ali, Akeel Abed Yasseen, Arafat Al-Dujailli, Ahmed Jafar Al-Karaqully, Katherine Ann McAllister, Alaa Salah Jumaah.

**Writing – review & editing:** Katherine Ann McAllister.

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
