## [Decision Letter · Decision Letter 0]

29 Nov 2021

PONE-D-21-31116The oxytocin receptor gene polymorphism rs2268491 and serum oxytocin alterations are indicative of autism spectrum disorder: 

A case-control paediatric study in Iraq with personalized medicine implicationsPLOS ONE

Dear Dr. McAllister,

Thank you for submitting your manuscript to PLOS ONE. After careful consideration, we feel that it has merit but does not fully meet PLOS ONE’s publication criteria as it currently stands. Therefore, we invite you to submit a revised version of the manuscript that addresses the points raised during the review process. Specifically, the authors should address the following points:Criterion 3. “Experiments, statistics, and other analyses are performed to a high technical standard and are described in sufficient detail”. Experiments must have been conducted rigorously, with appropriate controls and replication. Sample sizes must be large enough to produce robust results, where applicable. Methods and reagents must be described in sufficient detail for another researcher to reproduce the experiments described. (1) Authors should clarify sampling method and justify sample size. (2) Define severity of ASD. (3) Provide required details for all used kits and reagents, such as manufacturer, city, and country. (4) Experimental details on Isolation of genomic DNA and Agarose gel electrophoresis are commonly used techniques so that they should be summarized in the main text and the technical details can be better provided as supplementary. (5) Did the authors test data for normality? (6) Did the author consider confounders “e.g., the Median age of control is lower than that of ASD, which may affect serum OXT level”. (7) In table 2, authors should describe which statistical tests were used; moreover, data are presented as mean (SD) but some appear to be not normally distributed. (8) In addition “The ANOVA test found statistically significant difference (p=0.032) in OXT levels according to severity of ASD, as shown in Table 2”. Please, specify the difference? (9) In table 2, it seems that Oxytocin levels are lower in severe ASD (higher in mild ASD), which may be associated with older age. (10) It is interesting to see that healthy controls had highest OXT levels in the CC genotype, while ASD subjects had the highest OXT in the TT genotype; it would be meaningful to explain/discuss such findings.Criterion 4. “Conclusions are presented in an appropriate fashion and are supported by the data.” Authors have to revise the conclusion of their study to be based on study findings (e.g., “The measurement of peripheral OXT levels and OXTR genetic alterations provide a simple dual-biomarker test to gauge social functioning in the ASD patient for diagnostic, monitoring and support purposes”; however, this study didn’t investigate monitoring and support aspects). Other suggested implications/recommendations based on study findings/conclusion should be separated from the conclusion itself.  Criterion 5. “The article is presented in an intelligible fashion and is written in standard English.” (1) Table 1: better to show ASD and controls in columns and variables as rows. Please, provide other important data, such as gender and comorbid conditions. (2) The article contain several errors related to English language, spelling, and grammar.  *PLOS ONE *does not copyedit accepted manuscripts, so the language in submitted articles must be clear, correct, and unambiguous. We may reject papers that do not meet these standards. If the language of a paper is difficult to understand or includes many errors, we may recommend that authors seek independent editorial help before submitting a revision.Criterion 6. “The research meets all applicable standards for the ethics of experimentation and research integrity.”. The authors repeat the ethics statement twice, once under study populations “Formal parental consent was obtained from families who expressed interest in the study, who were provided with consent sheets and a questionnaire to complete” and under Ethical approval “All patients were informed about the purpose of the work, and a written consent obtained from each patient.”. Authors should provide this information once. Moreover, did they obtain consent from the patients “children?”, parents, or both?Criterion 7. “The article adheres to appropriate reporting guidelines and community standards for data availability.” (1) In the abstract and discussion, authors seem to selectively report some findings in an inappropriate manner, such as “study characteristics in the ASD population revealed a high level of consanguinity (36.66%)” although the percentage of consanguinity was higher among control group (40%). (2) The authors stated that “all data are fully available without restriction” but they did not provide unidentified patients’ data. Authors are required to make all data underlying the findings described fully available, without restriction, and from the time of publication. PLOS allows rare exceptions to address legal and ethical concerns, but this has to be completely explained. Please, review PLOS Data Policy.

We look forward to receiving your revised manuscript.

Kind regards,

Elsayed Abdelkreem, MD, PhD

Academic Editor

PLOS ONE

Journal Requirements:

Reviewers' comments:

Reviewer's Responses to Questions

**Comments to the Author**

1. Is the manuscript technically sound, and do the data support the conclusions?

Reviewer #1: Yes

2. Has the statistical analysis been performed appropriately and rigorously? 

Reviewer #1: I Don't Know

3. Have the authors made all data underlying the findings in their manuscript fully available?

Reviewer #1: Yes

4. Is the manuscript presented in an intelligible fashion and written in standard English?

Reviewer #1: Yes

5. Review Comments to the Author

Reviewer #1: This is an important study from clinical point of view. The whole article seemed to be written in readers friendly language, well articulated and intelligently structured. It is publishable, though I would like to suggest for further evaluation of statistical issues by an expert in this field.

6. PLOS authors have the option to publish the peer review history of their article (what does this mean?). If published, this will include your full peer review and any attached files.

Reviewer #1: No

---

## [Author Response · Author response to Decision Letter 0]

19 Jan 2022

1. Criterion 3:

a. (1) Authors should clarify sampling method and justify sample size.

Answer: (lines 126 paragraph manuscript) literature searches were performed to assess the prevalence of autism spectrum disorder. An estimated 1 % of the world’s population has autism spectrum disorder, with recent figures of 1.5%. Since the prevalence of the disease is a relatively uncommon event at 1 out of 100 cases, a case control study was selected as the most efficient sampling method to investigate autism among the Iraqi Arab population. A study sample size of N=120 (60 ASD and 60 controls) was feasible to manage in the face of opposition to the study from local tribal and religious restrictions. Our study had a 3:1 male to female ratio in the ASD and control populations. The DSM-5 states that “autism spectrum disorder is diagnosed four times more often in males than in females.”(American Psychiatric Association 2013), and most recent research suggest that it is closer to 3:1 (Loomes, Hull, and Mandy 2017). Therefore the study population is reflective of the gender ratio in ASD and provides a useful model to explore specific genetic associations.

b. (2) Define severity of ASD.

Answer: This was delineated in the PLOS PDF lines 145-182 as follow:

The severity was defined according to the Childhood Autism Rating Scale as follow:

The patients with ASD were stratified according to severity into the following sub-groups (20): mild ASD (n=39), moderate ASD (n=13) and severe ASD (n=8) according to the Childhood Autism Rating Scale 

(CARS) (21). The CARS is a diagnostic assessment method that rated individuals on a scale ranging from normal to severe and yields a composite score ranging from non-autistic to mildly autistic, moderately autistic, or severely autistic. The scale was used to observe and subjectively rate the following fifteen items in both ASD and case control subjects. Each of the following 15-items were scored in patients according to seven levels of severity by interview:

1. Relationship to people

2. Imitation

3. Emotional response

4. Body

5. Object use

6. Adaptation to change

7. Visual response

8. Listening response

9. Taste-smell-touch response and use

10. Fear and nervousness

11. Verbal communication

12. Non-verbal communication

13. Activity level

14. Level and consistency of intellectual response

15. General impressions

c. (3) Provide required details for all used kits and reagents, such as manufacturer, city, and country.

Answer: done. These were delineated in supplementary Table S1, S2

S1 Table. Instruments and Equipment used in the study. 

S2 Table. Chemicals and reagents for DNA extraction and polymerase chain reaction. 

d. (4) Experimental details on Isolation of genomic DNA and Agarose gel electrophoresis are commonly used techniques so that they should be summarized in the main text and the technical details can be better provided as supplementary. 

Answer: 

The experimental details for both isolation of genomic DNA and preparation of gels for electrophoresis are now summarized in the main manuscript with full lab protocols described in supplementary table 3.

e. (5) Did the authors test data for normality?

Answers: The Kolmogorov-Smirnov test was used to test normality, which showed a normal distribution, methods (statistical analysis): paragraph 270. The test for all variables was greater than 0.05 so that samples are normally distributed (Table 2).

f. Did the author consider confounders “e.g., the Median age of control is lower than that of ASD, which may affect serum OXT level”.

Answer. Ages were matched between the ASD and control group. There were no significant differences in age distribution between studied and control group P=0.708 (Table 2).

g. (7) In table 2, authors should describe which statistical tests were used; moreover, data are presented as mean (SD) but some appear to be not normally distributed.

Answer: done. Analysis of Variance (ANOVA) test were used for statistical analysis as the data were normally distributed which was ensured by doing Kolmogorov-Smirnov test (Table 2).

h. (8) In addition “The ANOVA test found statistically significant difference (p=0.032) in OXT levels according to severity of ASD, as shown in Table 2”. Please, specify the difference?

Answer: done. The oxytocin level was 165.02±28.03 pg/mL in mild cases, 152.78±27.86 pg/mL in moderate cases, and 129.09±11.04 pg/mL in severe cases. Furthermore, there were significant differences in oxytocin level when comparing mild and severe cases (p=0.001), moderate and severe cases (p=0.034). On other hand there were no significant differences in oxytocin level between mild and moderate cases (p=0.178) (Table 2, table footnotes).

i. (9) In table 2, it seems that Oxytocin levels are lower in severe ASD (higher in mild ASD), which may be associated with older age.

Answer. Done. Oxytocin level was higher in mild cases (165.02±28.03 pg/mL) than severe cases (129.09±11.04 pg/mL). However there was no significant correlation between age of the patient and oxytocin level as stated in the manuscript in lines 554 and Supplementary Figure 1 (P= 0.396, R=-0.140, R2=0.0196; Supplementary Fig 1). 

j. (10) It is interesting to see that healthy controls had highest OXT levels in the CC genotype, while ASD subjects had the highest OXT in the TT genotype; it would be meaningful to explain/discuss such findings.

Answer. Done (Discussion, main manuscript). In stratified ASD patients, serum oxytocin levels were higher in the mild category with the polymorphism genotype TT. This suggests that patients with milder ASD symptoms and the TT genotype may have evolved an ability to upregulate OXT levels to help or try to compensate for the cerebral social response skills deficiency caused by the OXTR polymorphism. The higher amounts of OXT will help to improve facial processing and human interpersonal contact. As showed in previous study D LoParo et al 2015 showed oxytocin receptor gene rs2268491 (‘T’ allele is risk-inducing) in the meta-analysis, these SNPs were shown to be substantially related with ASD, implying that signals from these SNPs may represent a shared connection with ASD. Liu x, et al. 2010 showed two SNPs (rs2268491 and rs2254298), which were significantly associated with ASD in there study in which re 2268491 with allele C predominant.

2. Criterion 4. “Conclusions are presented in an appropriate fashion and are supported by the data.” Authors have to revise the conclusion of their study to be based on study findings (e.g., “The measurement of peripheral OXT levels and OXTR genetic alterations provide a simple dual-biomarker test to gauge social functioning in the ASD patient for diagnostic, monitoring and support purposes”; however, this study didn’t investigate monitoring and support aspects). Other suggested implications/recommendations based on study findings/conclusion should be separated from the conclusion itself. 

Answer: Done. The final conclusion paragraph has been amended, the implications/recommendations of the study have been presented as such from lines 485

3. Criterion 5. “The article is presented in an intelligible fashion and is written in standard English.”

a. (1) Table 1: better to show ASD and controls in columns and variables as rows. Please, provide other important data, such as gender and comorbid conditions.

Answer. Done. Table 1 was corrected as requested by the reviewers. 

b. (2) The article contain several errors related to English language, spelling, and grammar. PLOS ONE does not copyedit accepted manuscripts, so the language in submitted articles must be clear, correct, and unambiguous. We may reject papers that do not meet these standards. If the language of a paper is difficult to understand or includes many errors, we may recommend that authors seek independent editorial help before submitting a revision.

Done. 

4. Criterion 6. “The research meets all applicable standards for the ethics of experimentation and research integrity.”. The authors repeat the ethics statement twice, once under study populations “Formal parental consent was obtained from families who expressed interest in the study, who were provided with consent sheets and a questionnaire to complete” and under Ethical approval “All patients were informed about the purpose of the work, and a written consent obtained from each patient.”. Authors should provide this information once. Moreover, did they obtain consent from the patients “children?”, parents, or both?

Answer: In fact, written consent was taken from the father of the ASD child only who is in charge and responsible for any matter regarding the ASD patients according to the societal customs in Iraq. The ethics statement has been updated as recommended. Please refer to paragraph lines 272

Criterion 7. “The article adheres to appropriate reporting guidelines and community standards for data availability.”

a. (1) In the abstract and discussion, authors seem to selectively report some findings in an inappropriate manner, such as “study characteristics in the ASD population revealed a high level of consanguinity (36.66%)” although the percentage of consanguinity was higher among control group (40%).

Answer: The sample size might be behind this variation which shows that the percentage of consanguinity was higher among control group. Much larger sample sizes will facilitate a clear picture on these variations. The consanguinity findings are now fully reported in the results and discussion sections lines 282 and 403: 

• Results: The rate of consanguinity among the Arabic study populations was high at 36.66% and 40% in the ASD and healthy control groups (Table 1).

• Discussion: Given that the consanguinity rate worldwide is 10.4% (7), the rates observed in both the ASD and control study populations are at least three fold higher.

b. (2) The authors stated that “all data are fully available without restriction” but they did not provide unidentified patients’ data. Authors are required to make all data underlying the findings described fully available, without restriction, and from the time of publication. PLOS allows rare exceptions to address legal and ethical concerns, but this has to be completely explained. Please, review PLOS Data Policy.

Done. All data is available as supplementary material as Supplementary S1 Appendix. 

5. Data sharing statement.

Answer: All the data cannot be made publically available as a results of local institution policy. However, it can be made available on a reasonable request from the corresponding author or from the department of pathology and forensic medicine (third party). 

Contact information of the department of pathology and forensic medicine is: Email: pathology.med@uokufa.edu.iq

(email included in the cover letter)

6. Please amend either the title on the online submission form (via Edit Submission) or the title in the manuscript so that they are identical.

Amended

7. PLOS PDF (phrase data not shown).

Done. This phrase has been removed, and the OXT and patient age Supplementary Fig 1 is now included.

---

## [Decision Letter · Decision Letter 1]

2 Feb 2022

PONE-D-21-31116R1The oxytocin receptor gene polymorphism rs2268491 and serum oxytocin alterations are indicative of autism spectrum disorder: A case-control paediatric study in Iraq with personalized medicine implications

PLOS ONE

Dear Dr. McAllister,

Thank you for submitting your manuscript to PLOS ONE. After careful consideration, we feel that it has merit but does not fully meet PLOS ONE’s publication criteria as it currently stands. Therefore, we invite you to submit a revised version of the manuscript that addresses the points raised during the review process.

Besides the important reviewers' comments shown below, authors are requiried to address the following points:

(1) Previous comment: Authors should clarify sampling method and justify sample size.

Authors’ response: (lines 126 paragraph manuscript) literature searches were performed to assess the prevalence of autism spectrum disorder. An estimated 1 % of the world’s population has autism spectrum disorder, with recent figures of 1.5%. Since the prevalence of the disease is a relatively uncommon event at 1 out of 100 cases, a case control study was selected as the most efficient sampling method to investigate autism among the Iraqi Arab population. A study sample size of N=120 (60 ASD and 60 controls) was feasible to manage in the face of opposition to the study from local tribal and religious restrictions. Our study had a 3:1 male to female ratio in the ASD and control populations. The DSM-5 states that “autism spectrum disorder is diagnosed four times more often in males than in females.”(American Psychiatric Association 2013), and most recent research suggest that it is closer to 3:1 (Loomes, Hull, and Mandy 2017). Therefore the study population is reflective of the gender ratio in ASD and provides a useful model to explore specific genetic associations.

New comment: Please, provide calculation of the sample size that would be required for this study. If the number enrolled in this study satisfy the calculated sample size, this would be perfect. If not, you can acknowledge this as one of the study limitations and provide justifications, including feasibility issues.

(2) Previous comment: Did the author consider confounders “e.g., the Median age of control is lower than that of ASD, which may affect serum OXT level”.

Authors response. Ages were matched between the ASD and control group. There were no significant differences in age distribution between studied and control group P=0.708 (Table 2).

New comment: Thank you for your important clarification. However, as shown in Table 2, the mean age for controls is lower than that of patients, and the p-value may become significant if larger number was included. This underscores the importance of sample size calculation.

(3) Previous comment: It is interesting to see that healthy controls had highest OXT levels in the CC genotype, while ASD subjects had the highest OXT in the TT genotype; it would be meaningful to explain/discuss such findings.

Authors’ response. Done (Discussion, main manuscript). In stratified ASD patients, serum oxytocin levels were higher in the mild category with the polymorphism genotype TT. This suggests that patients with milder ASD symptoms and the TT genotype may have evolved an ability to upregulate OXT levels to help or try to compensate for the cerebral social response skills deficiency caused by the OXTR polymorphism. The higher amounts of OXT will help to improve facial processing and human interpersonal contact. As showed in previous study D LoParo et al 2015 showed oxytocin receptor gene rs2268491 (‘T’ allele is risk-inducing) in the meta-analysis, these SNPs were shown to be substantially related with ASD, implying that signals from these SNPs may represent a shared connection with ASD. Liu x, et al. 2010 showed two SNPs (rs2268491 and rs2254298), which were significantly associated with ASD in there study in which re 2268491 with allele C predominant.

New comment: Thank you for these important discussions/thoughts . However, authors are encouraged to expand the discussion of this point in the main text (please, integrate the rest of your response, including the references, in the main text).

(4) Previous comment: In the abstract and discussion, authors seem to selectively report some findings in an inappropriate manner, such as “study characteristics in the ASD population revealed a high level of consanguinity (36.66%)” although the percentage of consanguinity was higher among control group (40%).

Authors’ response: The sample size might be behind this variation which shows that the percentage of consanguinity was higher among control group. Much larger sample sizes will facilitate a clear picture on these variations. The consanguinity findings are now fully reported in the results and discussion sections lines 282 and 403. •Results: The rate of consanguinity among the Arabic study populations was high at 36.66% and 40% in the ASD and healthy control groups (Table 1). •Discussion: Given that the consanguinity rate worldwide is 10.4% (7), the rates observed in both the ASD and control study populations are at least three fold higher

New comment: Thank you for your important clarifications. Please, integrate your comments regarding sample size in the main text.

(5)Other comments

Please, provide the age of study participants in table 1 (with other characteristics) rather than table 2.  It is not necessary to provide the p-value for the Kolmogorov-Smirnov test in table 2. Instead, just report this in the table footnote or results section.Authors are encouraged to acknowledge and discuss study limitations and generalizability of findings.  

We look forward to receiving your revised manuscript.

Kind regards,

Elsayed Abdelkreem, MD, PhD

Academic Editor

PLOS ONE

Journal Requirements:

Reviewers' comments:

Reviewer's Responses to Questions

**Comments to the Author**

1. If the authors have adequately addressed your comments raised in a previous round of review and you feel that this manuscript is now acceptable for publication, you may indicate that here to bypass the “Comments to the Author” section, enter your conflict of interest statement in the “Confidential to Editor” section, and submit your "Accept" recommendation.

Reviewer #2: All comments have been addressed

2. Is the manuscript technically sound, and do the data support the conclusions?

Reviewer #2: Partly

3. Has the statistical analysis been performed appropriately and rigorously? 

Reviewer #2: Yes

4. Have the authors made all data underlying the findings in their manuscript fully available?

Reviewer #2: Yes

5. Is the manuscript presented in an intelligible fashion and written in standard English?

Reviewer #2: Yes

6. Review Comments to the Author

Reviewer #2: Title: The oxytocin receptor gene polymorphism rs2268491 and serum oxytocin alterations are indicative of autism spectrum disorder: A case-control paediatric study in Iraq with personalized medicine implications

The manuscript is precisely written and it presents an excellent point that might help in the suggestion of personalized precision intervention strategy. I recommend the acceptance but only after major revision. From my experience the number of samples is satisfactory, however,

- Most recent prevalence of ASD should be cited:

E.g.: Bougeard Clémence, Picarel-Blanchot Françoise, Schmid Ramona, Campbell Rosanne, Buitelaar Jan. Prevalence of Autism Spectrum Disorder and Co-morbidities in Children and Adolescents: A Systematic Literature Review, Frontiers in Psychiatry, 12, 2021 DOI=10.3389/fpsyt.2021.744709

- The significantly high oxytocin in mild ASD patients compared to moderate and severe should be explained and supported by other studies if possible.

- Individuals with autism do not outgrow autism (This sentence is not clear please clarify or delete)

“Ignoring early and accurate diagnosis of this disorder might lead to secondary disorders such as depression and anxiety (Reference is needed). Please see the reference below:

Hollocks MJ, Lerh JW, Magiati I, Meiser-Stedman R, Brugha TS. Anxiety and depression in adults with autism spectrum disorder: a systematic review and meta-analysis. Psychol Med. 2019 Mar;49(4):559-572. doi: 10.1017/S0033291718002283. Epub 2018 Sep 4. PMID: 30178724.

- “These results suggest the ROC curve could become the gold standard for the identification of parameters that are sensitive and specific enough to support ASD diagnosis”. ROC curves are already known as excellent statistical tool in the field of biomarkers, so this statement should be corrected.

- The authors stated that “ within the stratified ASD population in this study, the highest OXT levels occurred in the mild subgroup, while the lowest OXT levels occurred in the severe group. This should be explained and supported at least with the fact that nasal oxytocin is recommended to decrease the severity of ASD symptoms (Support is mandatory).

- The significant difference between mild, moderate and severe ASD should be clearly presented in the table not only in the text. Roc curves for mild, moderate and severe should be presented independently.

- The authors mentioned that “This suggests that patients with milder ASD symptoms and the TT genotype may have evolved an ability to upregulate OXT levels to help or try to compensate for the cerebral social response skills deficiency caused by the OXTR polymorphism. The higher amounts of OXT will help to improve facial processing and human interpersonal contact” (Again support your suggestion).

- Please go through the manuscript below it might help

https://www.nature.com/articles/s41598-020-79109-0.pdf

Horiai, M., Otsuka, A., Hidema, S. et al. Targeting oxytocin receptor (Oxtr)-expressing neurons in the lateral septum to restore social novelty in autism spectrum disorder mouse models. Sci Rep 10, 22173 (2020). https://doi.org/10.1038/s41598-020-79109-0

Gene expression analysis shows that Oxt mRNA is up-regulated in brain and bone which indicate that Oxt is adaptive and important in restoring the homeostasis of the body.

7. PLOS authors have the option to publish the peer review history of their article (what does this mean?). If published, this will include your full peer review and any attached files.

Reviewer #2: **Yes: **Afaf El-Ansary

---

## [Author Response · Author response to Decision Letter 1]

7 Feb 2022

Response to Reviewers

PLOS Editor specific revisions:

The authors would like to thank the PLOS editor for these helpful manuscript suggestions.

• Previous comment: Authors should clarify sampling method and justify sample size.

New comment to address: Please, provide calculation of the sample size that would be required for this study. If the number enrolled in this study satisfy the calculated sample size, this would be perfect. If not, you can acknowledge this as one of the study limitations and provide justifications, including feasibility issues.

Response from authors: 

Methods, Lines (131-144) A sample size of N=36 was calculated as the minimal requirement to determine serum oxytocin alterations in the study cohort. The study variable of interest (peripheral oxytocin alterations) was identified as dichotomous (proportion). The sample size was calculated using the equation: N = 4 * Zα2 * p (1 − p)/w2. Where Zα is the confidence level, W is the width of the confidence interval and P is the study estimate of the proportion to be measured. The proportion of ASD patients who present with oxytocin alterations compared to controls was estimated at about 90% (S1 Appendix) with a selected confidence interval of ± 10 [16]

N = 4 * 1.96 * 0.9 (1 − 0.9)/0.22 = 17.64 (18 cases of ASD)

Minimum sample size for 1:1 case-control study cohort: N=36 

Please note that a bigger sample size is ideal for estimating a single OXT SNP in a population (about N=250), this is discussed in the new study limitation section of the discussion (lines 508 paragraph).

• Previous comment: Did the author consider confounders “e.g., the Median age of control is lower than that of ASD, which may affect serum OXT level”.

New comment to address: Thank you for your important clarification. However, as shown in Table 2, the mean age for controls is lower than that of patients, and the p-value may become significant if larger number was included. This underscores the importance of sample size calculation

Responses from authors: Thank you for raising the issue of confounders. The study is of a case control age and gender matched design which should reduce effects of these confounder variables. Although the mean age of patients with ASD is slightly higher - it is not significant (Table 1). The sample size calculations indicates that the study numbers are sufficient to investigate the variable of serum oxytocin. 

• Previous comment: It is interesting to see that healthy controls had highest OXT levels in the CC genotype, while ASD subjects had the highest OXT in the TT genotype; it would be meaningful to explain/discuss such findings. 

New comment: Thank you for these important discussions/thoughts. However, authors are encouraged to expand the discussion of this point in the main text (please, integrate the rest of your response, including the references, in the main text).

Responses from authors: The discussion of this key finding of the how OXT levels can differ in relation to OXTR genotype has been reworked and expanded in relation to social functioning with references (lines 465 to 488) 

• New comment: Thank you for your important clarifications. Please, integrate your comments regarding sample size in the main text.

Response: Done. Refer to the study limitation section of the discussion (lines 508 paragraph).

• Please, provide the age of study participants in table 1 (with other characteristics) rather than table 2. 

Response: Done

• It is not necessary to provide the p-value for the Kolmogorov-Smirnov test in table 2. Instead, just report this in the table footnote or results section.

Response: Done

• Authors are encouraged to acknowledge and discuss study limitations and generalizability of findings. 

Response: done. (Lines 507-519) ‘Ideally the conclusions presented in this study should be generalizable to the wider Iraqi Arab population. Therefore the study samples must be representative of the population and adequate in number (33). Feasibility issues in the Middle and South Euphrates population region limited recruitment of large sample sizes in the current study. Although the estimated study sample size was sufficient for the serum OXT analysis, numbers are still inadequate to generalize to the entire population. For example, consanguinity was high among the ASD populations as expected, but also the control group. The sample size might explain why the consanguinity proportions are higher among control patients. Sample size also affects the association between SNP markers and disease. Testing a single SNP marker in a 1:1 case control study requires 248 cases to achieve 80% statistical power according to the allelic genetic model, under certain assumptions (34). Insufficient study numbers are a limitation of the current investigation of SNP rs2254298.

Reviewer 2:

The authors would like to thank reviewer 2 for these helpful revisions.

• Most recent prevalence of ASD should be cited:

E.g.: Bougeard Clémence, Picarel-Blanchot Françoise, Schmid Ramona, Campbell Rosanne, Buitelaar Jan. Prevalence of Autism Spectrum Disorder and Co-morbidities in Children and Adolescents: A Systematic Literature Review, Frontiers in Psychiatry, 12, 2021 DOI=10.3389/fpsyt.2021.744709

Response: This is an excellent suggestion, we have updated our introductory prevalence of ASD literature, citing ‘recent estimates of 1.70 and 1.85% in US children aged 4 and 8 years respectively’ (lines 81-82)

• The significantly high oxytocin in mild ASD patients compared to moderate and severe should be explained and supported by other studies if possible.

Response: Done. Please refer to discussion lines 441- 457

• The trend for elevated OXT in this study agrees with other studies (14-16). However, the use of OXT as a biomarker of ASD is not straightforward, as these results are contradictory with those reporting lower OXT in ASD patients (10-12). However, within the stratified ASD population in this study, the highest OXT levels occurred in the mild subgroup, while the lowest OXT levels occurred in the severe group. Our study found that ASD severity categories had statistically significant (p=0.032) difference in OXT levels. This trend does agree with the previous study of ASD children in Iraq (12). Here OXT level were highest in mild autistic patients with a significant decrease (p<0.05) in moderate autistic patients, and a highly significant decrease (p<0.01) in severe autistic patients compared with control. Interestingly, this variability in OXT levels in stratified ASD patients reported both Iraqi studies may relate to a key finding of the Stanford study (27). The US investigators noted that pretreatment blood OXT concentrations also predicted response to intranasal OXT treatment(27). Those individuals with the lowest pretreatment OXT concentrations showed the greatest social improvement (27). Therefore, the ASD severe patients with lowest OXT in our study could be suitable candidates for individualized intranasal therapy with OXT to boost their social functioning. Individuals with autism do not outgrow autism (This sentence is not clear please clarify or delete) 

Response: Done. Sentence is updated to: ‘Autism is a lifelong neurodevelopmental disorder’.

• “Ignoring early and accurate diagnosis of this disorder might lead to secondary disorders such as depression and anxiety (Reference is needed). Please see the reference below: Hollocks MJ, Lerh JW, Magiati I, Meiser-Stedman R, Brugha TS. Anxiety and depression in adults with autism spectrum disorder: a systematic review and meta-analysis. Psychol Med. 2019 Mar;49(4):559-572. doi: 10.1017/S0033291718002283. Epub 2018 Sep 4. PMID: 30178724.

Response: Done. Thank you for the reference suggestion.

• “These results suggest the ROC curve could become the gold standard for the identification of parameters that are sensitive and specific enough to support ASD diagnosis”. ROC curves are already known as excellent statistical tool in the field of biomarkers, so this statement should be corrected.

Response: Done. ROC curves are now discussed as an existing ‘gold standard statistical tool (lines 432)

• The authors stated that “within the stratified ASD population in this study, the highest OXT levels occurred in the mild subgroup, while the lowest OXT levels occurred in the severe group. This should be explained and supported at least with the fact that nasal oxytocin is recommended to decrease the severity of ASD symptoms (Support is mandatory).

Response: Done. (Lines 451-456). The US investigators noted that pretreatment blood OXT concentrations also predicted response to intranasal OXT treatment (27). Those individuals with the lowest pretreatment OXT concentrations showed the greatest social improvement (27). Therefore, the ASD severe patients with lowest OXT in our study could be suitable candidates for individualized intranasal therapy with OXT to boost their social functioning.

• The significant difference between mild, moderate and severe ASD should be clearly presented in the table not only in the text. Roc curves for mild, moderate and severe should be presented independently

Response: Done. Figure 1 has been updated to show all ROC curves independently, including mild, moderate and sever (Fig 1C-E).

• The authors mentioned that “This suggests that patients with milder ASD symptoms and the TT genotype may have evolved an ability to upregulate OXT levels to help or try to compensate for the cerebral social response skills deficiency caused by the OXTR polymorphism. The higher amounts of OXT will help to improve facial processing and human interpersonal contact” (Again support your suggestion).

Please go through the manuscript below it might help

https://www.nature.com/articles/s41598-020-79109-0.pdf

Horiai, M., Otsuka, A., Hidema, S. et al. Targeting oxytocin receptor (Oxtr)-expressing neurons in the lateral septum to restore social novelty in autism spectrum disorder mouse models. Sci Rep 10, 22173 (2020). https://doi.org/10.1038/s41598-020-79109-0

Gene expression analysis shows that Oxt mRNA is up-regulated in brain and bone which indicate that Oxt is adaptive and important in restoring the homeostasis of the body.

Response: Done. Suitable discussion material (lines 484-487) has been added from the suggested paper. ‘A recent mouse ASD model study demonstrated that targeting OXTR-expressing neurons in the lateral septum restores social skills (31). The results of the current study and previous research (27, 31) suggest that OXT is adaptive and can restore the homeostasis of the body.

---

## [Decision Letter · Decision Letter 2]

28 Feb 2022

The oxytocin receptor gene polymorphism rs2268491 and serum oxytocin alterations are indicative of autism spectrum disorder: A case-control paediatric study in Iraq with personalized medicine implications

PONE-D-21-31116R2

Dear Dr. McAllister,

We’re pleased to inform you that your manuscript has been judged scientifically suitable for publication and will be formally accepted for publication once it meets all outstanding technical requirements.

Kind regards,

Elsayed Abdelkreem, MD, PhD

Academic Editor

PLOS ONE

Additional Editor Comments (optional):

Reviewers' comments:

Reviewer's Responses to Questions

**Comments to the Author**

1. If the authors have adequately addressed your comments raised in a previous round of review and you feel that this manuscript is now acceptable for publication, you may indicate that here to bypass the “Comments to the Author” section, enter your conflict of interest statement in the “Confidential to Editor” section, and submit your "Accept" recommendation.

Reviewer #2: All comments have been addressed

2. Is the manuscript technically sound, and do the data support the conclusions?

Reviewer #2: Yes

3. Has the statistical analysis been performed appropriately and rigorously? 

Reviewer #2: Yes

4. Have the authors made all data underlying the findings in their manuscript fully available?

Reviewer #2: Yes

5. Is the manuscript presented in an intelligible fashion and written in standard English?

Reviewer #2: Yes

6. Review Comments to the Author

Reviewer #2: Thanks for your response to my previous comments

The authors answered my comments and use the provided references to modify the discussion accordingly and I think the manuscript is suitable for publication.

7. PLOS authors have the option to publish the peer review history of their article (what does this mean?). If published, this will include your full peer review and any attached files.

Reviewer #2: **Yes: **Afaf Kamal El-Din El-Ansary

---

## [Editor Report · Acceptance letter]

7 Mar 2022

PONE-D-21-31116R2 

The oxytocin receptor gene polymorphism rs2268491 and serum oxytocin alterations are indicative of autism spectrum disorder: A case-control paediatric study in Iraq with personalized medicine implications 

Dear Dr. McAllister:

I'm pleased to inform you that your manuscript has been deemed suitable for publication in PLOS ONE. Congratulations! Your manuscript is now with our production department. 

Kind regards, 

on behalf of

Dr. Elsayed Abdelkreem 

Academic Editor

PLOS ONE